# Exploring ethnic minority women's experiences of maternity care during the SARS-CoV-2 pandemic: a qualitative study

Jeeva Reeba John ![ORCID],[1] Gwenetta Curry,[2] Sarah Cunningham-Burley[1]

[1]Centre for Biomedicine, Self, and Society, The University of Edinburgh, Usher Institute of Population Health Sciences and Informatics, Edinburgh, UK
[2]The University of Edinburgh, Usher Institute of Population Health Sciences and Informatics, Edinburgh, UK

**Correspondence to**
Dr Jeeva Reeba John;
jeeva.john@ed.ac.uk

## ABSTRACT

**Objective** To explore the experiences of pregnancy, childbirth, antenatal and postnatal care in women belonging to ethnic minorities and to identify any specific challenges that these women faced during the SARS-CoV-2 pandemic.

**Design** This was a qualitative study using semistructured interviews of pregnant women or those who were 6 weeks postnatal from Black, Asian and minority ethnic backgrounds. The study included 16 women in a predominantly urban Scottish health board area.

**Results** The finding are presented in four themes: 'communication', 'interactions with healthcare professionals', 'racism' and 'the pandemic effect'. Each theme had relevant subthemes. 'Communication' encompassed respect, accent bias, language barrier and cultural dissonance; 'interactions with healthcare professionals': continuity of care, empathy, informed decision making and dissonance with other healthcare systems; 'racism' was deemed to be institutional, interpersonal or internalised; and 'the pandemic effect' consisted of isolation, psychological impact and barriers to access of care.

**Conclusions** This study provides insight into the specific challenges faced by ethnic minority women in pregnancy, which intersect with the unique problems posed by the ongoing SARS-CoV-2 pandemic to potentially widen existing ethnic disparities in maternal outcomes and experiences of maternity care.

## Strengths and limitations of this study

► This study addresses a gap in the literature by using qualitative methods to provide an in-depth exploration of minority ethnic women's experiences of maternity care during the SARS-CoV-2 pandemic.

► This study explores the perspectives of pregnant ethnic minority women at different stages of pregnancy, as well as the postnatal period.

► A flexible topic guide was developed using existing literature, in collaboration with an obstetrician, a social scientist and a race and ethnicity lecturer, bringing together different ideologies to a complex issue.

► The interviewer is a female obstetrician, and although none of the participants were known to the interviewer clinically, they were aware of the interviewer's role, and it is important to acknowledge that this could have had an indirect effect on participant responses.

## INTRODUCTION

While the overall maternal mortality rate (MMR) in the UK has remained relatively low over the past decade,[1] the gulf in maternal outcomes between white women and women from ethnic minority backgrounds continues to expand. The latest UK confidential enquiry into maternal deaths (MBRRACE-UK) report showed significant racial variations in maternal mortality. Black women were four times, while Asian women were two times as likely to die as White women during pregnancy, delivery or postpartum between 2016 and 2018.[2]

These results are comparable to the MMR in minoritised ethnic groups in the USA and Netherlands despite key differences in healthcare systems.[3 4] Inequalities in maternal outcomes are long standing and reproducible in other high-income countries, however the causes of these disparities remain unclear. Ethnic minorities form a significant proportion of some of the most disadvantaged groups in the UK facing intersecting social determinants of health including unemployment and deprivation.[5 6] Although some studies have demonstrated correlation between these social elements and poor maternal outcomes, they do not fully explain the gap.[7 8]

Previous studies have investigated the experiences of pregnant immigrant women accessing maternal care in the UK. Women who had negative experiences with healthcare professionals avoided accessing maternal services.[9] Unsurprisingly, not engaging with maternal care services can negatively impact both maternal and fetal health, thereby increasing the risk of severe morbidity and

mortality. On the contrary, women from ethnic minority backgrounds who did wish to engage experienced limitations in accessing maternal care services in the UK, which also led to poorer health outcomes.[10] In countrywide surveys of maternity care, ethnic minority women were more likely to report poor patient experience; in some cases, attributed to stereotyping and racism.[11 12] Experiences of racism and discrimination have also been linked to poor outcomes for minoritised populations.[13]

The SARS-CoV-2 pandemic has not only shone a spotlight on these disparities but may have exacerbated them.[14] A key component in establishing equality of maternal healthcare provision is the examination of women's experiences of these services. There is little qualitative research on the lived experiences of all ethnic minority women during or immediately following pregnancy in the UK. Previous qualitative literature in the UK has tended to focus on nuanced aspects of maternal care,[15] particular health conditions during pregnancy[16] or specific ethnic groups.[17] This study aimed to explore the experiences of pregnancy, childbirth, antenatal and postnatal care in all women belonging to ethnic minority communities and to identify any specific challenges that these women faced during the SARS-CoV-2 pandemic.

## METHODS
### Participant characteristics and recruitment
Women who were pregnant or within 6 months of delivery from a Black, Asian and minority ethnic background were recruited for the current study. Sixteen women were recruited: seven Black African, one Black Caribbean, three Asian-Indian, one Asian-Chinese, one Asian-Bangladeshi, one Asian-Pakistani and two Arab. Nine of the women were antenatal and seven were postnatal. All but one of the participants had themselves been born outside the UK.

This study was performed in all of the community midwifery hubs and obstetric units within a predominantly urban Scottish health board. A health board is a regional authority in Scotland with local responsibility for the delivery of health services. During the SARS-CoV-2 pandemic, changes to delivery of maternal healthcare within this health board followed national and Royal College of Obstetricians and Gynaecologists' guidance and included a universal shift from face-to-face consultations to telephone clinics and restriction of visitors during hospital admission during scan appointments and following delivery.[18] Units were provided with study recruitment packs containing a participant information sheet, consent form and prepaid envelope. Midwives were encouraged to share study information with ethnic minority patients and to provide them with recruitment packs. With their consent, participant details were securely passed onto researcher, JRJ, who telephoned each patient 48 hours after receiving a recruitment pack. Participants who could not speak English were provided appropriate translations of study documents and interpreters were

made available for interviews. All participants confirmed consent prior to each interview. The interview topic guide was developed by all three authors and was used to aid semistructured interviews carried out by JRJ.

### Data collection
Data collection occurred between December 2020 and January 2021. Audio-recorded interviews were transcribed verbatim by First Class Secretarial Services. Collected data was kept confidential by allocating a distinct code to each woman to protect anonymity. Data was collected until no new themes were identified and inductive and deductive thematic saturation was achieved.

### Data analysis
This study adopted a data analysis methodology based on thematic analysis.[19] The transcriptions were read and re-read, a coding frame was constructed and the data coded to identify initial themes. A qualitative interpretive approach was taken, and thematic analysis was conducted with continuous consultation between researchers JRJ and GC. A consensus discussion regarding theme development was undertaken by all three authors.

### Patient and public involvement
There was no formal PPI group that assisted with this study, and transcripts were not returned to participants for comment. However, in the course of recruitment and interviews, advice from participants and healthcare professionals was sought and used to create a flexible interview topic guide. A summary of study findings will be available on the Centre for Biomedicine, Self and Society for general access, and study findings will also be disseminated among study participants with consent having been obtained at the time of interview.

## RESULTS
Sixteen women participated in telephone interviews, which were up to an hour in duration (see table 1 for sample characteristics). No follow-up interviews were carried out.

Four principal themes were identified from interview data. These included communication, interactions with healthcare professionals, racism and the pandemic effect. Each theme had subthemes, as presented in figure 1.

Direct quotes relating to each theme are presented in boxes 1–4.

### Communication
#### Respect
Participants who were satisfied with the maternal health system made reference to respectful communication. This was particularly important during episodes of high maternal stress (box 1: *Respect,* P15). Participants concurred that community midwives and health visitors were considerate in their routine communication (box 1: *Respect,* P1). A significant majority of the group conversely

**Table 1** Study participant characteristics

| | N (%) |
|---|---|
| **Ethnicity** | |
| Black-African | 7 (43.7) |
| Black-Caribbean | 1 (6.3) |
| Asian-Indian | 3 (18.7) |
| Asian-Chinese | 1 (6.3) |
| Asian-Bangladeshi | 1 (6.3) |
| Asian-Pakistani | 1 (6.3) |
| Arab | 2 (12.5) |
| **Age (years)** | |
| 25–29 | 1 (6.3) |
| 30–34 | 10 (62.5) |
| 35–40 | 4 (24.9) |
| >41 | 1 (6.3) |
| **Religion** | |
| Christian | 6 (37.5) |
| Muslim | 7 (43.7) |
| Hindu | 1 (6.3) |
| Atheist | 2 (12.5) |
| **Country of birth** | |
| UK | 1 (6.2) |
| Outside UK | 15 (93.8) |
| **Year of immigration** | |
| N/A as born within UK | 1 (6.3) |
| 2000 to 2009 | 3 (18.7) |
| 2010 to 2019 | 11 (68.7) |
| 2020 | 1 (6.3) |
| **Parity** | |
| Nulliparous | 2 (12.5) |
| One | 6 (37.5) |
| Two | 6 (37.5) |
| Three | 2 (12.5) |
| **Antenatal/postnatal** | |
| Antenatal | 9 (56.3) |
| Postnatal | 7 (43.7) |
| **Comorbidities** | |
| Yes | 2 (12.5) |
| No | 14 (87.5) |
| **First languages** | |
| Fulani | 2 (12.5) |
| Yoruba | 2 (12.5) |
| Arabic | 1 (6.3) |
| Bengali | 1 (6.3) |
| English | 1 (6.3) |
| Farsi | 1 (6.3) |
| French | 1 (6.3) |
| | Continued |

**Table 1** Continued

| | N (%) |
|---|---|
| Hindi | 1 (6.3) |
| Igala | 1 (6.3) |
| Jamaican Patois | 1 (6.3) |
| Mandarin | 1 (6.3) |
| Spanish | 1 (6.3) |
| Telagu | 1 (6.3) |
| Urdu | 1 (6.3) |
| **Employment** | |
| Yes | 13 (81.3) |
| No | 3 (18.7) |

felt that they received disrespectful care during unsolicited hospital visits (box 1: *Respect*, P11; P16).

### Accent bias

All participants except one were first-generation immigrants with median duration of stay in the UK of 9 years. The majority of the cohort had non 'British' accents and identified bias due to accent as being a significant concern, sometimes perceived to impede access to emergency care, and prevent equality of maternal care received. One participant highlighted having a non 'Westernised' accent as being interpreted as a proxy for lower socioeconomic status and educational attainment and that this was a unique barrier with regards to telephone consultations (box 1: *Accent bias*, P12).

### Language barrier

In this study, most participants spoke English fluently; only one required an interpreter. Despite the high standard of English spoken, most participants felt that language barriers were the most common cause of miscommunication between themselves and healthcare professionals. They concurrently felt they themselves were more likely to make inappropriate decisions regarding their healthcare as a result of misinterpretation (box 1: *Language barrier*, P8). Healthcare professional misunderstanding due to language barriers was also described as a prime feature in barriers to effective communication (box 1: *Language barrier*, P2; P9). In contrast, the participant who required an Arabic interpreter had not experienced any instances of miscommunication when an interpreter was present during preorganised appointments. However, she admitted facing major communication challenges during unscheduled calls when an interpreter was not available. She identified a gap in recognition from healthcare professionals regarding this particular scenario and reported that she often resorts to searching the internet for pregnancy advice even during emergencies due to anxiety surrounding communication (box 1: *Language barrier*, P14).

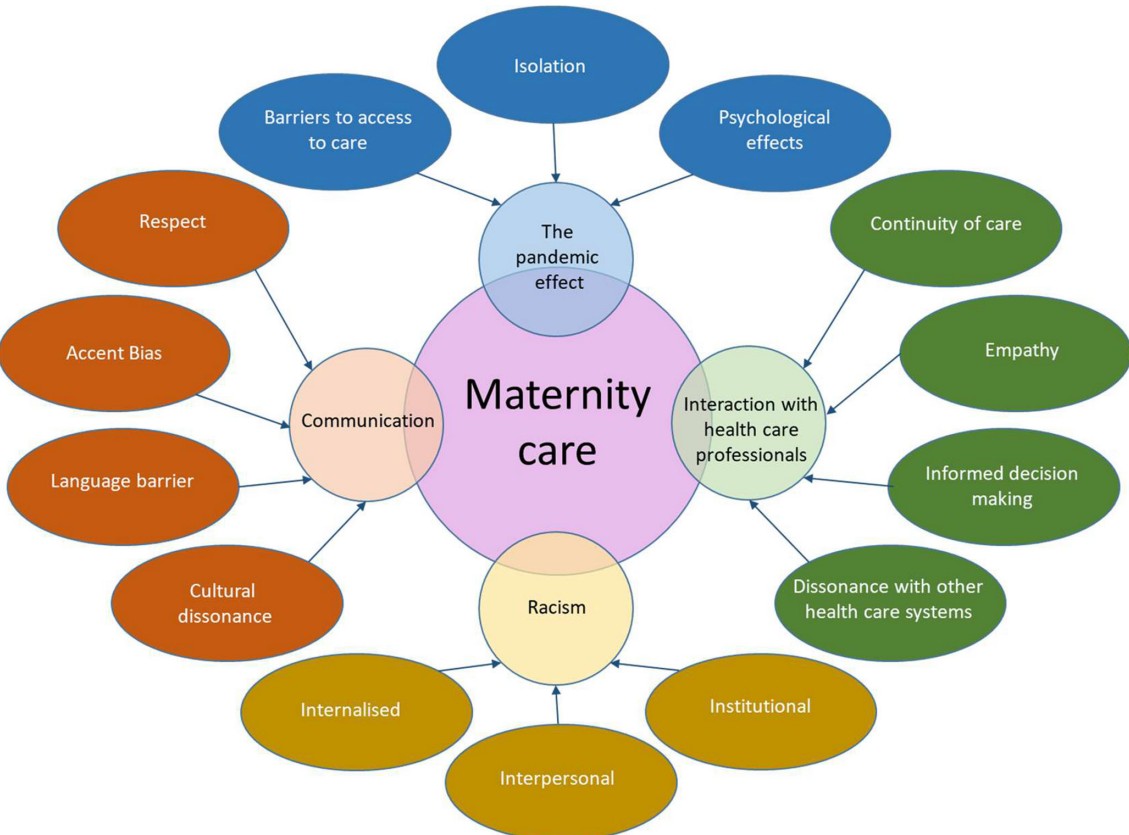

**Figure 1** Themes and subthemes.

## Cultural dissonance

Cultural dissonance was identified as another significant barrier to effective communication. Cultural dissonance between participant and healthcare professional (box 1: *Cultural dissonance*, P2) as well as between participants and their wider community (box 1: *Cultural* dissonance, P8) both impacted quality of communication equally. This could be in the form of religious discordance in routine clinical practice (box 1: *Cultural* dissonance, P11) or misunderstanding of wider cultural context (box 1: *Cultural dissonance*, P5).

## Interactions with healthcare professionals

Quality of interaction with healthcare professionals was generally associated with four key domains: continuity of care, empathy, informed decision making and dissonance with previous experiences of maternity care. This theme, although strongly interlinked with the previous one, was derived separately to emphasise the non-verbal and institutional aspects of healthcare interactions that affected participants' experiences of maternity care.

## Continuity of care

Overall, participants felt they received good continuity of care throughout their pregnancy. Primiparous women particularly valued routine postnatal check-ups (box 2: *Continuity of care*, P2). A common sentiment that arose in women requiring regular input from secondary care during the antenatal period was ineffective communication between their community midwives and hospital midwives or obstetricians and vice versa, sometimes resulting in omission of crucial clinical information (box 2: *Continuity of care*, P7). In the women who experienced prolonged hospital admissions, the inability to discuss ongoing care with the same healthcare professionals led to varying information being given to the patient leading to dissolution of trust (box 2: *Continuity of care*, P3).

## Empathy

The most important aspect of interactions with healthcare professionals was the presence of empathy. In instances where the healthcare professional was deemed indifferent, participants often cited being hurried, feeling unheard and uncared for, which negatively impacted the interaction, and the overall perception of maternity care (box 2: *Empathy*, P3). Participants who experienced apathy in previous pregnancies recognised that the possibility of recurrence was a source of anxiety throughout subsequent pregnancies (box 2: *Empathy*, P9; P12).

## Informed decision making

Although all participants reported some level of information provision from healthcare providers regarding clinical decision making, almost everyone agreed that they would benefit from more thorough discussions. Most participants received information about their pregnancy in the form of signposting to books or websites but they

## Box 1 Direct quotations relating to theme of communication

**Communication**
*Respect*
P1 I never feel like I am not in my country.
P6 I could sense that she was busy and she could have rather been off, you know, not having that one extra person popping into triage at that point. For me, it was just a case of, well, you know, I need this, so I can only apologise, but there's nowhere else I can go.
P11 I will also want some respect as well, because I'm, I'm capable to getting their respect. So, if other people win respect then I have to make the confidence well yes, I'm capable of getting their respect.
P15 I had a CVS at eleven weeks and during the procedure, I was kind of, um, emotional. Um, I was being talked to by the nurses and the midwives very well. I really appreciated the way they spoke to me.

*Accent bias*
P12 I don't call about everything. But when you have to call through, you're not taken seriously once your accent is heard. That's just the truth, yeah. I just feel that the…there…there's a little bit of language when it comes to…once your accent is heard, you…you…you are not the privileged lot, let me put it that way. So nobody's gonna appoint you an appointment. They don't…they don't think it's necessary

*Language barrier*
P8 So, therefore, if I'm given the same sort of information at the same level…with someone that is English speaking, the person that is English speaking will understand a lot better…and probably make better decisions than I would.
P2 It creates like a…a gap in communication where if something you express is not clearly understood so maybe they could be left with some misinterpretation
P9 Um, you know, when I initially came my English was not good. Eh, and it was harsh for me, the…with the way they communicated…because I don't know the language and they know; they know that I don't know the language. And I'm already depressed because I don't know whether my baby will survive or not.
P14 So if she feels confident to speak about the baby, to phone and speak about the issue, she, she will…yeah, she will do it. But if she feels that the person will not…she's phoning will not understand her because she won't be able to deal with the message or to say whatever she want to say, she prefers not to phone and to explore other solutions. It's really difficult sometimes…eh, to know what to do or who to speak to.

*Cultural dissonance*
P2 how your culture is too…can…can be a factor as well. 'Cause, umm, this can…this can cause a disparity where…a misinterpretation again where…where in my culture this is okay…But…and…and how I express myself, it might come off as you're not being accepted but, umm, it doesn't mean…it does not come off from a negativity on my side 'cause it's…it's my culture……but because someone else is from a different culture and they clash, there…there comes a…a misinterpretation, so that can affect care as well
P5 Um, and then the circumcision part of it is just on hold at the moment. Um, and, and that's…that's something that's quite important for us to get kind of done straightaway
P8 …and as a Nigerian I think people just think, a Caesarean section, no no no no,…because that has happened. So the first…the first thing I had when I have a planned C-section like, oh my gosh, have you tried? Don't you know God can heal you?

Continued

## Box 1 Continued

P11 One thing people, sometimes I find some people don't like, so well, I don't want to show people that I'm Muslim. There are lots of times because of Muslim, because of the bomb blast and other sorts of things, they are pointing me.

expressed that their individual information needs would have been better met by one-to-one discussions (box 2: *Informed decision making*: P6). Women who felt that their questions remained unanswered did not feel involved in shared decision making (box 2: *Informed decision making:* P3).

### Dissonance with other healthcare systems
Fifty per cent of the study population had experienced maternity care outside the present health board, with a significant proportion having delivered outside the U.K. All participants who previously delivered outside the U.K. had experienced privatised healthcare.

Access to early pregnancy care and shared decision making were some of the reasons provided by women who previously delivered in other parts of the U.K. reporting higher satisfaction with the study health board in the current pregnancy. The majority of the participants who had delivered abroad felt that their previous pregnancy experiences were generally better compared with care received in this health board. Reasons given for this discordance varied among individual participants and included improved access to ultrasound scans, availability of medical specialists and treatment by healthcare providers (box 2: *Dissonance with other healthcare systems*, P12; P14). A high proportion of primiparous women had sought advice from relatives, friends and/or healthcare professionals in their country of birth (box 2: *Dissonance with other healthcare systems*, P3).

### Racism
#### Institutional
Institutional racism was highlighted as a significant issue in pregnancy care by most of the participants. A small proportion of Black participants concomitantly worked in healthcare and were able to provide examples of racial discrimination against themselves from their colleagues (box 3: *Institutional*, P10). Some of the participants expressed distrust in maternal healthcare due to their concerns that medical research and treatments are tailored for their White counterparts (box 3: *Institutional*, P8). Participants who had experienced institutional racism were also likely to perceive barriers to accessing healthcare compared with White women (box 3: *Institutional*, P12).

#### Interpersonal
Most women gave examples of acts of racism against their friends, or family rather than themselves (box 3: *Interpersonal*, P6; P12).

## Box 2 Direct quotations relating to theme of interaction with healthcare professionals

### Interactions with healthcare professionals

*Continuity of care*

P2 So the continuity of care both in and outside of appointments, so it was just quite tremendous, just really good. …that has impressed me as well, and, umm, it's…it's just the thoroughness of it, to be honest, so your journey is not left at the hospital.

P3 And then every time a different doctor would come with a different approach……and they were not really clear and explaining to me what's going on, so I was very stressed. Um, and I really wanted to see a consultant, but I had to wait for that for quite a long time. Kind of I had to cry actually to. And I wasn't relying on the system anymore because I was hearing different opinions every time.

P7 So maybe they thought that I was still seen by my midwives in my, in my GP surgery or something, and they were…they never checked the size of your, um, tummy and they did not really check the position of the baby and all those kind of things.

*Empathy*

P3 …and the midwives were very busy, um, so they couldn't attend me. And at one point, I just asked the midwife to, to look after my baby while I go to the toilet, and when I came back, she was not there and my baby had, like, vomited all over her face. So it was really, really the worst time of my life

P9 But the only thing what I felt was they haven't paid enough…much empathy for a human being. Eh, what I felt most was they didn't care enough to care for a human being. It was just like a product, that's what I felt. I was just like a product over there. There was no value for the emotions.

P12 I had a horrible…horrible experience with the…the midwives… midwives on…on duty. Okay, I've had two…I had two kids but I had no complication and so having the third one and I was, err…I…I had to go through an emergency Caesarean and all that it was very traumatic for me as a person and…but I don't think they understood that.

*Informed decision making*

P3 but it was mainly up to me as well to read and ask questions…rather than getting, um, like information.

P6 I've been given the Ready, Steady, Baby book and stuff…so there is reading materials and links and online antenatal classes, but I wouldn't say there's been that much support in terms of, you know, actually sitting me down and talking to me and explaining different things, it's been more, this is where you can go and read on stuff if you want.

*Dissonance with other health care systems*

P3 I would go and I would come back and she was shocked every time because the process in Iran is really different. I was wishing that I was actually in Iran so I could get more help from specialists.

P12 As much as you have to pay to get, umm, your baby delivered in America…they…they have a lot more care than in the United Kingdom.

P14 In Jordan she used to go every month to the doctor to check everything about the baby. She really wish if, whenever she goes to see the midwife, there is something to assure her about the baby's health…

## Box 3 Direct quotations relating to theme of racism

### Racism

*Institutional*

P8 Treatment for black people will be different from treatment for white people.

P10 Um, they are white, they don't want to talk to you.

P12 Most of the time for me to get care for my children, I have to say it with a very loud voice or sounding like the child is just they're going to die in a minute before I'm responded to.

*Interpersonal*

P6 Um, and she said that no, this is not right, you know, um, she'd kind of gone to the GP and stuff, and she'd spoken to the midwives and they were like, oh, it's nothing that serious, but at the same time, there was white women who were having the same issue, but being taken more seriously upon.

P10 Yeah, because they didn't listen to us anyway. I think they listen to their people than, you know, us, and they didn't have the patience. Because I remember when I had my baby, and then I had a C-section when I was in the hospital, I was so much in pain when I called one of the midwives to come and help me to feed my baby, she was proper shouting, oh, you need to try it for yourself, you know. Just not…just not being polite. Talking to me, you know, so rude.

P12 in the United Kingdom they act as if you are privileged to be…so I'm…I'm also talking as, err, B…BAME, right? Like a black woman, right? They act like you're privileged that you are not having a baby in…in the bush.

*Internalised*

P6 so I've got a bit more I guess, a cold-headed, um, kind of, I guess, but I'm quite direct, and it's only made me, like hearing all these cases and stories and, you know, the women's experience, it only kind of makes me a bit more, I guess, hard skinned, um, just to make sure that, you know, I need to make sure that no-one is overlooking me, just for the sake of my skin.

P7 Well, I believe that we receive equal healthcare regardless of the colour of the skin. But what I think, the body, the nature of the body or the gene is different in compared to the people here. Our gene or the nature of our body and whatever it is quite different… Um, I, I think our bodies are much more weak or something like that maybe. Maybe less immune or less resistant, or maybe like that.

P11 It's not just only country, my culture, my, my view, not only religion, because we are third world country…I'm not from the developed country. When…if I'm coming from like Canada or the US or something I can, I can show that confidence

White women (box 3: *Internalised*, P6). In another case, a participant felt that she did not have the right to question the healthcare she was receiving, being originally from a low-income country (box 3: *Internalised,* P11). Yet, another participant strongly believed that disparities in maternal health outcomes could be explained by physiological differences between ethnic groups (box 3: *Internalised*, P7).

### The pandemic effect

This study was developed in order to better understand the lived experiences of women from ethnic minority backgrounds during maternity care. Due to the time period that the study was conducted in, the SARS-CoV-2

### Internalised

In participants who expressed internalised racist beliefs, these almost always affected health behaviours. In one example, a significant delay in diagnosis of a common condition led one participant to believe that the main problem was her lack of assertiveness compared with

Box 4   Direct quotations relating to the pandemic effect

**The pandemic effect**

*Isolation*

P2 So COVID kind of, umm, robbed me of that experience which I'm used to back at home, where I used to attend my clinics…So clinics gave you that…that…that, umm, avenue of socialisation, meeting another mothers, new mothers……creating, umm, friendships…and…and also, you know, a continuance after birth

P3 I'm not getting enough support for taking care of my baby, she's crying day and night, and I couldn't have my husband or anyone because of COVID

P5 I had to go to all my appointments on my own.

*Psychological impact*

P11 COVID actually is really stressful mentally…you know. I can't go anywhere, nobody can come, which is in a pregnant woman, uh, isn't…I'd love to go somewhere, but we couldn't go anywhere because of the restriction.

P13 I found it very hard when you're coming to the country without knowing anyone and the coronavirus, everyone is…doing lockdown so it was very…difficult, I was very depressed. I was very anxious, yeah, umm, I feel worried a lot.

*Barriers to access of care*

P5 I think there were disadvantages, you don't have that face to face contact.

P6 I don't know if that's the pandemic causing people to be a bit more relaxed and, you know, like okay, here's the leaflet, you can go and do the reading.

P10 Because when I was explaining about my breathing, they didn't want to see me, they were saying that they think that I'm having a coronavirus…and I really forced, yeah, I said I'm not having a coronavirus, it's like this is… I know that this is related to my cardiomyopathy. I have really, really struggled, I can say that. I've been calling, calling, talking, talking, they don't want to see me.

P11 But they said, no you can't come, you have to send a picture and you have to email us. Which is sometimes, you know, is not easy to do with the phone. Because I'm not a doctor.

P14 she can't phone the GP to describe what is the problem. And because of the Corona, she understands that it's not easy to get an appointment. So she tried to read on the internet how she can help herself.

pandemic and its direct and indirect consequences were naturally brought up by all participants. The subthemes below illustrate the most common impacts of the pandemic as mentioned by the participants.

## Isolation

Most women reported feeling isolated during their pregnancy due to features specific to the SARS-CoV-2 pandemic. This was particularly a problem for those who felt that they would have benefitted from the presence of a companion when important information relating to their pregnancy was being relayed to them (box 4: *Isolation*, P5). Although most women were understanding of the limitations posed by the pandemic, a significant proportion expressed loneliness exacerbated by not being able to engage with their usual pregnancy support networks,

and did not feel that virtual groups mitigated this effect (box 4: *Isolation,* P2; P3).

## Psychological impact

The detrimental effect of the pandemic on mental health during pregnancy was highlighted by all participants. Women expressed feeling more anxious, fearful and lacking autonomy in comparison with their previous pregnancies. This sentiment was particularly emphasised by the participants who had little or no family members nearby (box 4: *Psychological impact*, P11; P13). The majority of participants reported that they were not routinely asked about their mental health in relation to the pandemic by healthcare professionals. Fewer still were signposted to appropriate support groups for help.

## Barriers to access of care

Most participants had some of their routine face-to-face appointments replaced by telephone calls. The majority felt that virtual appointments were not as effective, especially during unscheduled care (box 4: *Barriers to access of care*, P5). Women expressed uncertainty surrounding the accuracy of information relayed via telephone (box 4: *Barriers to access of care*, P6). In circumstances where participants did seek physical consultations, they experienced barriers and often had to repeatedly call in order to be seen (box 4: *Barriers to access of care*, P10; P11). On the contrary, some women delayed seeking medical help due to apprehensions surrounding contracting SARS-CoV-2 (box 4: *Barriers to access of care*, P14).

## DISCUSSION
### Main findings
This study focused on the lived experiences during the SARS-CoV-2 pandemic of minority ethnic women who were pregnant or had delivered within 6 weeks prior to interview. There were four emergent themes including communication, interactions with healthcare professionals, racism and the effect of the pandemic, with further subthemes identified. Although many of the issues identified are not unique to minority ethnic women, the findings emphasise previous study results.[20–22] The systemic inadequacies highlighted in maternity care provision for women from ethnic minority backgrounds have been exacerbated by the health service modifications resulting from the SARS-CoV-2 pandemic.

### Strengths and limitations
This is the first qualitative study, to our knowledge, which explores the maternity experiences of women from ethnic minority backgrounds during the SARS-CoV-2 pandemic in the U.K.

This study is methodologically limited by reliance on narratives, which are often subject to recall bias, from a small number of patients at a single Scottish health board and may therefore restrict generalisability of results. However, we believe that our inclusive selection criteria

provided a degree of homogeneity in participant experiences, which was helpful in facilitating rich discussion. Moreover, delivery within the same health board provided a foundation from which varying experiences could be compared and contrasted.

In addition, we acknowledge that although ethnic differences are important study factors, they are inevitably intertwined with socioeconomic factors. Ethnic minority groups with a higher socioeconomic position are possibly over-represented in this study. Selection bias cannot be excluded as a higher socioeconomic status is associated with increased utilisation of maternity care services, which is how participant recruitment took place. Thus, views expressed by participants in this study are likely to be under-representative of the true challenges in accessing maternal healthcare faced by minority ethnic pregnant women from lower socioeconomic backgrounds in other parts of the U.K.

The interviewer, JRJ, is an obstetrician, and although none of the participants were known to JRJ clinically, they were aware of the interviewer's role, and it is important to acknowledge that this could have had an indirect effect on participant responses. Finally, the research team's different positionalities in clinical and public health backgrounds may have influenced data interpretation. However, we believe that this diversity broadened and deepened our understanding of these women's experiences.

## Interpretations

The proportion of births to migrant women has increased annually in line with growing immigration, from 11.6% in 1990[23] to 28.2% in 2019.[24] Adequate and appropriate healthcare provision for ethnic minority groups has long been recognised as integral to safe maternity care.[25] The Equality Act 2010 states that NHS treatment and care, including maternal healthcare, should be equitable and no person should be discriminated against on the basis of their ethnicity.[26] Despite recognition of issues specific to this cohort, more than a decade later, UK maternity care services have failed to respond to increasing levels of diversity, and as a result, ethnic minority women have continued to suffer from worse pregnancy outcomes. The Right to Health document decrees that the four key elements of universal right to healthcare are that governments must ensure that healthcare services are ethically and culturally acceptable to all, accessible to all, available in sufficient quantity and of good quality.[27] This study provides evidence to support that development of new and innovative strategies is urgently required to guarantee that all ethnic minority women receive culturally acceptable, accessible and equitable maternal healthcare in the UK not only to tackle existing disparities but also to combat the additional detrimental effects of the SARS-CoV-2 pandemic.

The issues raised are not exclusive to ethnic minority women; however, it is plausible that the need for support, effective communication and good quality care are not met to a greater degree in this cohort as reported in previous studies[28][29] and that these long-standing issues are exacerbated to a larger extent among these women as a result of the SARS-CoV-2 pandemic.

Good communication forms the foundation of good clinical care, and therefore, it is unsurprising that issues surrounding different aspects of communication were identified as a key theme. It is striking that although the majority of this group were fluent in English, they still identified it as a contributing problem, which mirrors previous studies' conclusions that language proficiency does not always facilitate a good pregnancy experience.[30][31] These issues are likely to be amplified with the changes in maternal care provision during the SARS-CoV-2 pandemic, such as the predominance of telephone consultations, increasing risk of misunderstanding and misinformation. Communication, or lack thereof, played a major role in participants' perceptions of whether they were receiving acceptable care. This consisted of routine or emergency interactions with midwives, obstetricians, general practitioners and health visitors. The majority of participants reported that regular communication with their community midwives and health visitors was excellent, however, collective areas of improvement were suggested for dealings with secondary care staff during emergency visits. This requires maternity services to engage with local communities and stakeholder groups to better understand heterogeneous socio-cultural needs and to augment staff cultural competency.

Despite overlap, interactions with healthcare professionals is treated as a separate theme to communication to highlight the role of the healthcare professional in providing maternal care beyond delivery of information. A unique subtheme that arose in this study is the effect that previous maternal care outside the UK has on the perception of maternal health services during the current pregnancy. Women who had not delivered elsewhere were still likely to discuss their care with friends, family or healthcare professionals from their country of birth where practices differed. The discordance in maternal healthcare proved a source of worry and remained unaddressed by their healthcare providers. A broader understanding of variations in maternal care is vital to provide reassurance. Although not specific to ethnic minority women, the problems of continuity of care between primary and secondary care identified in this study are likely to be aggravated by the changes posed by the SARS-CoV-2 pandemic. Logistical problems such as inadequate or absent interpretation services or short appointment times negatively impact the relationships formed between minoritised patients and healthcare professionals and represent modifiable factors influencing the holistic nature of maternity care.

As a general trend, longer duration of time spent in the UK was associated with awareness of racism impacting maternal health outcomes and personal experience of racial discrimination, which is in contrast to previous findings that found that women who came to the UK more

recently had a more negative perception of their care than women who had been in the UK for longer.[32] Sadly, the majority of participants were able to narrate examples of their friends, family members or wider community who had experienced racial discrimination both within and out with the context of healthcare in the UK. Most women reported personal health behaviours that they had developed in response to others' experiences of discrimination. Previous studies have implicated patients' own experiences of racism in poor maternal health,[33] however, the influence of health behaviour modification as a consequence of cognisance of discrimination that others face, as highlighted by this study, is not well explored. Previous findings show that women exposed to high levels of racism may be at increased risk of adverse maternal health outcomes.[34]

All women recruited to this study had experienced a significant portion of their maternity care during lockdown restrictions due to the SARS-CoV-2 pandemic. Only two women in this study were primiparous so the majority of participants were able to contrast their experience during this pregnancy to previous pregnancies prior to the pandemic. It is important to give due consideration to the specific challenges faced by ethnic minority women during this period such as exacerbation of communication issues and increased barriers to accessing necessary care.

## CONCLUSION

Maternal health outcome inequalities experienced by ethnic minority groups are multifactorial in nature. With the increasingly diverse pregnant population within the UK, tackling these discrepancies must be a priority. The first three themes reported in this study offer plausible causes to known differences in MMR of minoritised women in the UK compared with white women. This study provides insight into the specific challenges faced by these groups in pregnancy, which intersect with the unique problems posed by the ongoing SARS-CoV-2 pandemic to potentially widen existing ethnic disparities in maternal outcomes and experiences of maternity care. Future research should focus on in-depth exploration of maternity systems to inform the development of effective and robust interventions with the aim of reducing ethnicity-based maternal health inequalities.

**Acknowledgements** We would like to thank all of the women that participated in this study and acknowledge the help of obstetricians and midwives who assisted with recruitment.

**Contributors** JRJ was responsible for the conception of the study, planning, delivery, qualitative interviews, analysis of the study and wrote the first draft of the paper. GC and SC-B contributed to the planning of the study, interpretation of results and provided critical feedback on the draft paper. All authors read and approved the final manuscript.

**Funding** This work was supported by Wellcome Trust grant number 209519/Z/17/Z.

**Competing interests** None declared.

**Patient and public involvement** Patients and/or the public were not involved in the design, or conduct, or reporting, or dissemination plans of this research.

**Patient consent for publication** Not required.

**Ethics approval** The study was approved by the Research Ethics Committee of West of Scotland (20/WS/0168).

**Provenance and peer review** Not commissioned; externally peer reviewed.

**Data availability statement** All data relevant to the study are included in the article or uploaded as supplementary information.

**ORCID iD**
Jeeva Reeba John http://orcid.org/0000-0001-9651-4193

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
