## [Reviewer comments · BMJ Open]

ARTICLE DETAILS

TITLE (PROVISIONAL)	Exploring ethnic minority women's experiences of maternity care during the SARS-CoV-2 pandemic: a qualitative study
AUTHORS	John, Jeeva; Curry, Gwenetta; Cunningham-Burley, Sarah

VERSION 1 – REVIEW

REVIEWER	Evans, Catrin University of Nottingham, School of Health Sciences,
REVIEW RETURNED	01-Apr-2021

GENERAL COMMENTS	This study is on a very important topic. However, there is so little detail regarding the context, study design, philosophy, methodology and methods that I feel it does not meet the required standard. The study is presented in an unusual way, leading me to believe that the author is not very familiar with qualitative research. For a future submission, I would suggest that the authors follow the COREQ reporting guidance for qualitative studies. There is a vast body of literature around racism and ethnic minority women's experiences of maternity care in the UK. This does not feature at all in the Introduction. The contribution of this research to the wider body of knowledge in this area therefore is very hard to assess and is not made explicit. Overall, the detail on methodology/methods is sparse. The data is highly descriptive and poorly theorised or otherwise contextualised. This greatly limits its utility or transferability. Likewise, the themes feel highly descriptive and generic and it is not very obvious how these issues have, or have not, been impacted by Covid (which is the purported aim of the study). An assessment of the value of this study is made more difficult by the fact that there is almost no detail on the study context or location – very little description of existing service design or, especially, the service modifications made during covid. The study therefore currently seems to relate very specifically to one health board with insufficient detail or literature presented to assess its wider transferability or value. Whilst this is such an important topic and I commend the authors for their enquiries into this area, I strongly suggest that the paper needs considerable further work – (i) situating it in the context of the wider literature in this area and (ii) providing a far more rigorous account of the methodology and methods – before publication could be considered
--

REVIEWER	McKinn, Shannon The University of Sydney, Sydney School of Public Health
REVIEW RETURNED	11-Apr-2021

GENERAL COMMENTS	Thank you for the chance to review this paper, which is clearly written and deals with an important and interesting topic. I have some suggestions that I hope you will agree will strengthen the manuscript for publication. Abstract: 1. Remove section “Main outcome measures” – inappropriate to study design (could move additional info up to “Objective” Intro: 2. The introduction is very short and although the authors state there is little qualitative research on the experiences of ethnic minority women, there is some relevant qual and mixed methods research (e.g. with immigrant women - https://bmjopen.bmj.com/content/bmjopen/9/12/e029478.full.pdf) and I think it needs to be engaged with here. Methods: 3. Just a note - there is a strange compression of lines at the beginning of the methods section in the version provided for review – I’m not sure if I have access to the whole Methods section – the first thing I have in my version is the subheading “Participant characteristics and recruitment.” There is info in the abstract about participant inclusion criteria (pregnant or up to 6 weeks post-partum) which I can’t see in the Methods, so I might be missing a section? 4. Was self-identification as being part of an ethnic minority group the only criteria used for purposive sampling? I’m not sure that I would call this process (as you have described it) purposive sampling as the wider sample is “ethnic minority women who are pregnant or recently gave birth”, so I’m not really sure where the “purposive” part comes into it. Did you purposively sample to include participants from a broad range of ethnic minority groups, for example? Or for a range of English language proficiency? 5. I’m not familiar with the term “Health board” – could you explain what this means for readers outside of Scotland/UK? 6. Could you expand on your description of your analysis process? Your COREQ list claims an inductive coding process (themes derived from the data), however your final themes correspond quite strongly with the topics in your interview guide, so would it be fair to say there was more of a combination of deductive and inductive coding? 7. You stated that GC acted as an external validator of the analysis process. What did this external validation entail? Given that coding was performed by one researcher (JJ), I assume it wasn’t a double-coding process, so I am interested to know how this was done, and why it was deemed to be necessary, given that validity is a construct that is generally applied to quantitative rather than qualitative research. Results:
---

8. I would like more explanation of how the themes of “communication” and “interactions with health professionals” are sufficiently thematically distinct so as to justify two separate themes, as they seem highly interrelated and it’s not really clear to me how they are different.

9. “Half of the study population had experienced maternity care out with the study health board” – I’m not sure what this means.

10. Section on dissonance with other health care systems: You state that women who had delivered in other parts of the UK were more satisfied with the current health board; and that women who had delivered overseas were less satisfied with their current care. This contrast is not sufficiently teased out in this paragraph, and it’s not clear in which context the more (perceived) “positive” aspects of care mentioned in this paragraph (access to early care, more access to specialists, more access to U/S) were available to them. It was clearer in the quotes table, but not everyone will read these unfortunately!

11. Was there any interaction between the first three themes and the context of COVID-19? Given the objectives of the study, the bulk of the results seem to be decontextualized from the pandemic.

Discussion

12. This section is lacking substantial engagement with the literature on the experiences of ethnic minority women in maternity care and needs to be significantly rewritten in order to be suitable for publication, in my opinion. I don’t purport to be 100% across this area of research in the UK, but I suspect (confirmed by a quick google search) that there is quite a bit that hasn’t been engaged with here (including recent literature – the most recent paper cited in this discussion is from 2013). There has also been a lot of work done in the US that I think you would find relevant to your findings.

13. What further research is needed on this topic/based on your findings?

Conclusion

14. The conclusion is quite weak, and is almost just a restating of your research objective. What are the specific challenges that you found? Why should the reader care about them? You need a stronger ‘take-home message’ for the reader here.

Tables

15. If it doesn’t compromise the anonymity of your study participants, it would be useful to have some contextual characteristics attached to the quotes in the tables

VERSION 1 – AUTHOR RESPONSE

Reviewer 1

As per reviewer 1's comments, we have put more detail into the introduction with appropriate reference to existing literature in order to situate our article in the context of the wider literature in this area. We have also added more detail into the methods section as far as the word count allows us in doing so, and hope you find this a more rigorous account of the methods used. The restrictive word count has limited our ability to discuss theory/philosophy in great detail.

Reviewer 2:

Abstract:

1. Remove section "Main outcome measures" – inappropriate to study design (could move additional info up to "Objective")

Thank you for highlighting this. We have now amended this accordingly.

Intro:

2. The introduction is very short and although the authors state there is little qualitative research on the experiences of ethnic minority women, there is some relevant qual and mixed methods research (e.g. with immigrant women - <https://bmjopen.bmj.com/content/bmjopen/9/12/e029478.full.pdf>) and I think it needs to be engaged with here.

Thank you again, and we hope you find that we have engaged more with previous literature in the introduction now.

Methods:

3. Just a note - there is a strange compression of lines at the beginning of the methods section in the version provided for review – I'm not sure if I have access to the whole Methods section – the first thing I have in my version is the subheading "Participant characteristics and recruitment." There is info in the abstract about participant inclusion criteria (pregnant or up to 6 weeks post-partum) which I can't see in the Methods, so I might be missing a section?

The subheading "Participant characteristics and recruitment" is indeed the first thing in Methods. We have not noticed the compression of lines that you mention, sorry about this.

4. Was self-identification as being part of an ethnic minority group the only criteria used for purposive sampling? I'm not sure that I would call this process (as you have described it) purposive sampling as the wider sample is "ethnic minority women who are pregnant or recently gave birth", so I'm not really sure where the "purposive" part comes into it. Did you purposively sample to include participants from a broad range of ethnic minority groups, for example? Or for a range of English language proficiency?

We agree with your comment and have now removed purposive as a descriptor of the sampling method.

5. I'm not familiar with the term "Health board" – could you explain what this means for readers outside of Scotland/UK?

We have now provided a definition of "Health board" within the main text.

6. Could you expand on your description of your analysis process? Your COREQ list claims an inductive coding process (themes derived from the data), however your final themes correspond quite strongly with the topics in your interview guide, so would it be fair to say there was more of a combination of deductive and inductive coding?

That would be very fair to say, and it was a combination of deductive and inductive coding, and we have changed the text to reflect this.

7. You stated that GC acted as an external validator of the analysis process. What did this external

validation entail? Given that coding was performed by one researcher (JJ), I assume it wasn't a double-coding process, so I am interested to know how this was done, and why it was deemed to be necessary, given that validity is a construct that is generally applied to quantitative rather than qualitative research.

It was not a double coding process. Analysis was a dynamic process with continuous discussions between GC and JJ, and the text has been changed to reflect this.

Results:

8. I would like more explanation of how the themes of "communication" and "interactions with health professionals" are sufficiently thematically distinct so as to justify two separate themes, as they seem highly interrelated and it's not really clear to me how they are different.

This has now been addressed in the main text. From the discussions with the participants, non-verbal and institutional factors relating to interaction with health care professionals was a major theme, and felt to be a separate theme to that of communication, which related to language, accent, respectful communication, and cultural nuances of speech.

9. "Half of the study population had experienced maternity care out with the study health board" – I'm not sure what this means.

This sentence has now been reworded. 50 percent of the study population had had previous maternity care in settings not studied in the current body of work.

10. Section on dissonance with other health care systems: You state that women who had delivered in other parts of the UK were more satisfied with the current health board; and that women who had delivered overseas were less satisfied with their current care. This contrast is not sufficiently teased out in this paragraph, and it's not clear in which context the more (perceived) "positive" aspects of care mentioned in this paragraph (access to early care, more access to specialists, more access to U/S) were available to them. It was clearer in the quotes table, but not everyone will read these unfortunately!

We hope that you will find this paragraph to be clearer in its intention now.

11. Was there any interaction between the first three themes and the context of COVID-19? Given the objectives of the study, the bulk of the results seem to be decontextualized from the pandemic.

This was a deliberate attempt on our part to ensure that our target audience understood that most of the themes demonstrated in this study are valid and pertinent outside the context of the pandemic, and represent easily modifiable factors whilst engaging with patient from ethnic minority backgrounds. The section on the effect of the pandemic itself, and then the discussion further tease out the potential ways in which the pandemic has exacerbated pre-existing issues within this population.

Discussion

12. This section is lacking substantial engagement with the literature on the experiences of ethnic minority women in maternity care and needs to be significantly rewritten in order to be suitable for publication, in my opinion. I don't purport to be 100% across this area of research in the UK, but I suspect (confirmed by a quick google search) that there is quite a bit that hasn't been engaged with here (including recent literature – the most recent paper cited in this discussion is from 2013). There has also been a lot of work done in the US that I think you would find relevant to your findings.

We hope that you find more recent literature cited now within the discussion. Although, we agree that there is other qualitative literature, substantive qualitative literature searches within the U.K. show that previous studies have either focussed on a particular health condition affecting ethnic minorities or a specific ethnic group or a particular aspect of maternal health care as opposed to a broader exploration throughout the pregnancy course of women belonging to all ethnic minority groups accessing maternal care in the U.K.

13. What further research is needed on this topic/based on your findings?

We hope you find this question answered within the updated conclusion.

Conclusion

14. The conclusion is quite weak, and is almost just a restating of your research objective. What are the specific challenges that you found? Why should the reader care about them? You need a stronger 'take-home message' for the reader here.

We have rewritten the conclusion extensively taking into account your suggestions, and hope you find it suitable.

Tables

15. If it doesn't compromise the anonymity of your study participants, it would be useful to have some contextual characteristics attached to the quotes in the tables

We decided not to attach personal characteristics to the quotations in order to protect anonymity.

VERSION 2 – REVIEW

REVIEWER	McKinn, Shannon The University of Sydney, Sydney School of Public Health
REVIEW RETURNED	29-Jun-2021
GENERAL COMMENTS	Thank you for your revisions. I want to note that the marked up copy provided with this submission has substantial differences to the clean version of the manuscript. I have referred to the clean copy only for this review. I have only very minor additional points. 1. Introduction should be split into two or more paragraphs.2. In the section on 'Dissonance with other healthcare systems': I previously commented I wasn't entirely sure what you meant in the first sentence of this paragraph. While this is now clearer, I think the issue may actually be the phrase 'out with,' which a quick Google informs me is used only in Scottish English. To avoid confusion for other non-Scottish readers, I would change this to (the admittedly much less charming!) 'outside'

VERSION 2 – AUTHOR RESPONSE

Reviewer comments:

- "Introduction should be split into two or more paragraphs."

As per your recommendation, the introduction has now been divided into paragraphs.

- "In the section on 'Dissonance with other healthcare systems': I previously commented I wasn't entirely sure what you meant in the first sentence of this paragraph. While this is now clearer, I think the issue may actually be the phrase 'out with,' which a quick Google informs me is used only in Scottish English. To avoid confusion for other non-Scottish readers, I would change this to (the admittedly much less charming!) 'outside'"

Thank you for highlighting this, and we have now changed outwith to outside.